An updated understanding of Texas bumble bee (Hymenoptera: Apidae) species presence and potential distributions in Texas, USA

Beckham Jessica L. jessica.beckham@unt.edu 1
Atkinson Samuel 2
1 Advanced Environmental Research Institute, University of North Texas , Denton , TX , United States of America
2 Advanced Environmental Research Institute and Department of Biological Sciences, University of North Texas , Denton , TX , United States of America
Ward Darren
Electronic publication date: 2017 Aug 10
Publication date: 2017
Volume: 5
Electronic Location ID: e3612
Received 2017 Apr 17; Accepted 2017 Jul 5
Copyright: ©2017 Beckham and Atkinson
Copyright year: 2017
Copyright holder: Beckham and Atkinson
License: This is an open access article distributed under the terms of the Creative Commons Attribution License, which permits unrestricted use, distribution, reproduction and adaptation in any medium and for any purpose provided that it is properly attributed. For attribution, the original author(s), title, publication source (PeerJ) and either DOI or URL of the article must be cited.
License URL: https://creativecommons.org/licenses/by/4.0/

Keywords: Pollinators, Species decline, Conservation, Species distribution modeling, MaxEnt modeling, Natural history collections, Citizen science data, Bumble bees

Funding: Texas Parks and Wildlife Department Advanced Environmental Research Institute This work was supported by the Texas Parks and Wildlife Department (2015 State Wildlife Grant) and the Advanced Environmental Research Institute, University of North Texas. There was no additional external funding received for this study. The funders had no role in study design, data collection and analysis, decision to publish, or preparation of the manuscript.

==============================
Texas is the second largest state in the United States of America, and the largest state in the contiguous USA at nearly 700,000 sq. km. Several Texas bumble bee species have shown evidence of declines in portions of their continental ranges, and conservation initiatives targeting these species will be most effective if species distributions are well established. To date, statewide bumble bee distributions for Texas have been inferred primarily from specimen records housed in natural history collections. To improve upon these maps, and help inform conservation decisions, this research aimed to (1) update existing Texas bumble bee presence databases to include recent (2007–2016) data from citizen science repositories and targeted field studies, (2) model statewide species distributions of the most common bumble bee species in Texas using MaxEnt, and (3) identify conservation target areas for the state that are most likely to contain habitat suitable for multiple declining species. The resulting Texas bumble bee database is comprised of 3,580 records, to include previously compiled museum records dating from 1897, recent field survey data, and vetted records from citizen science repositories. These data yielded an updated state species list that includes 11 species, as well as species distribution models (SDMs) for the most common Texas bumble bee species, including two that have shown evidence of range-wide declines: B. fraternus (Smith, 1854) and B. pensylvanicus (DeGeer, 1773). Based on analyses of these models, we have identified conservation priority areas within the Texas Cross Timbers, Texas Blackland Prairies, and East Central Texas Plains ecoregions where suitable habitat for both B. fraternus and B. pensylvanicus are highly likely to co-occur.

Introduction

Bumble bees (Bombus spp.) are an important group of insect pollinators that provide sustaining pollination services for both agricultural systems and biological communities (Corbet, Williams & Osborne, 1991; Klein et al., 2007; Potts et al., 2010; Garibaldi et al., 2013). Unfortunately, bumble bees face many modern challenges, including those posed by pesticide use (Gill, Ramos-Rodriguez & Raine, 2012; Whitehorn et al., 2012; Rundlöf et al., 2015), climate change (Kerr et al., 2015), disease (Szabo et al., 2012), and habitat loss (Kearns, Inouye & Waser, 1998; Grixti et al., 2009; Winfree et al., 2009). Consequently, bumble bee losses have been noted across Europe (Carvell, 2002; Sarospataki, Novak & Viktoria, 2005; Goulson et al., 2006; Fitzpatrick et al., 2007; Kosior et al., 2007; Goulson, 2010), Asia (Yang, 1999; Matsumura, Yokoyama & Washitani, 2004; Inoue, Yokoyama & Washitani, 2008; Xie, Williams & Tang, 2008), and North America (Thorp, 2005; Thorp & Shepherd, 2005; Colla & Packer, 2008; Grixti et al., 2009; Colla & Ratti, 2010; Cameron et al., 2011; Colla et al., 2012). Furthermore, evidence presented by Bartomeus et al. (2013) suggests that bumble bees are among the most vulnerable of North American bee taxa.

Because bumble bees are the primary pollinators of several agricultural crops, declines in bumble bee populations may threaten the permanence of foods such as blueberries, tomatoes, and peppers (Hatfield et al., 2012; Shipp, Whitfield & Papadopoulos, 1994; Whittington & Winston, 2004). Additionally, research has indicated that the loss of bumble bees from wild ecosystems could result in a pronounced decline in overall plant diversity (Memmott, Waser & Price, 2004), limited seed production by native plants, and less effective pollination by other pollinators (Brosi & Briggs, 2013).

Historically, eight species of bumble bees have been documented in Texas (Franklin, 1913; Warriner, 2012): B. auricomus (Robertson, 1903), B. bimaculatus Cresson, 1863, B. fervidus (Fabricius, 1798), B. fraternus (Smith, 1854), B. griseocollis (DeGeer, 1773), B. impatiens Cresson, 1863, B. pensylvanicus (DeGeer, 1773), and B. variabilis (Cresson, 1872). Additionally, B. pensylvanicus sonorus, sometimes classified as the distinct species B. sonorus Say, 1837 (Franklin, 1913; Stephen, 1957; Thorp, Horning & Dunning, 1983; Warriner, 2012), has been documented in Texas. For the purposes of this research we have treated this taxon as a western morphological variant of B. pensylvanicus following Milliron (1973), Labougle (1990), Poole (1996), and Williams et al. (2014), and refer to it hereafter as B. pensylvanicus sonorus. However, it should be noted that the taxonomic status of this taxon is still debatable, and genetic studies are warranted to address this question.

Though bumble bee declines can be difficult to assess owing to a lack of long-term monitoring data (Berenbaum et al., 2007), some studies have suggested regional and national declines for several species whose ranges extend into Texas, including B. auricomus (Colla et al., 2012), B. fervidus (Colla & Packer, 2008; Colla, Richardson & Williams, 2011; Colla et al., 2012), B. fraternus (Colla & Packer, 2008; Grixti et al., 2009; Hatfield et al., 2014), B. pensylvanicus (Berenbaum et al., 2007; Colla & Packer, 2008; Grixti et al., 2009; Cameron et al., 2011; Colla, Richardson & Williams, 2011; Colla et al., 2012), and B. variabilis (Grixti et al., 2009; Colla, Richardson & Williams, 2011; Colla et al., 2012). To our knowledge, the only recent attempt to assess the current status of Texas bumble bee populations took place in a 24-county region of northeast Texas, where five species had historically been found (Beckham et al., 2016). In that study, the results of targeted field surveys (2010–2014) were compared to natural history specimen records. While the researchers did not reconfirm presence of the regionally rare B. bimaculatus or B. variabilis in northeast Texas, they showed that the current relative abundances of the potentially declining B. fraternus and B. pensylvanicus were equivalent to historic levels in the study region, as was that of the nationally stable B. griseocollis. Nevertheless, compelling evidence for bumble bee declines across North America has raised awareness of the need for conservation measures. In Texas this has led to three taxa (B. pensyvlanicus, B. pensylvanicus sonorus, and B. variabilis) being designated as species of greatest conservation need in the Texas Conservation Action Plan (Texas Parks and Wildlife Department, 2012).

An effective species conservation program requires delineation of the distribution of the species of interest (Eken et al., 2004), as well as an understanding of their habitat. Prior to the current study, efforts to establish Texas bumble bee distributions included the compilation and mapping of Texas specimen data from natural history collections by Warriner (2012) and targeted field studies in the northeastern portion of the state (Beckham et al., 2016). Williams et al. (2014) also modeled range-wide species distributions for North American bumble bees, to include Texas species. While informative and foundational, these studies left some question as to the fine-scale distributions of Texas bumble bee species, as well as gaps in data for portions of the state.

Unlike many insect taxa, bumble bees can often be identified to species in the field and in photographs, and so data collected by citizen scientists can provide invaluable information for bumble bee research. These data have been used to monitor populations and nesting densities (Osborne et al., 2008; Lye et al., 2012), and have served as presence data for species distribution models (SDMs) (Kadoya et al., 2009). In recent efforts to spread awareness and fill gaps in presence data, two citizen scientist projects were established specifically for Texas volunteers to submit their bumble bee sightings, in the form of photographs with locality data: the Texas Bumblebees Facebook page, founded in 2011, and the iNaturalist Bees and Wasps of Texas project, founded in 2014. These repositories now contain hundreds of sightings recorded by numerous volunteers from across the state that can be used as additional presence data for SDMs, complementing those provided by natural history collection records and field surveys.

SDMs estimate the geographic distribution of a species using field observations and associated environmental predictor variables. Over the last few decades, SDMs have become increasingly useful tools for conservation planning (Guisan & Thuiller, 2005). The Maximum Entropy (MaxEnt) algorithm is attractive because its implementation is straightforward and makes use of presence-only data, along with environmental variables associated with presence localities, to produce robust distribution models (Phillips, Dudík & Schapire, 2004; Phillips, Anderson & Schapire, 2006).

In recent years MaxEnt has successfully been used for a variety of applications related to bumble bees. To aid in conservation planning, Koch & Strange (2009) used historic specimen data to construct range maps for four North American species, and Penado, Rebelo & Goulson (2016) predicted the potential ranges of rare species in the Iberian Peninsula, also revealing the possibility of undiscovered populations in mountainous areas. Others have used MaxEnt to evaluate species trends, including Cameron et al. (2011), who assessed the rangewide persistence of eight North American species, and Dellicour et al. (2017), who inferred past and present distributions of European bumble bees to investigate range shifts. And, to aid in invasive species management decisions, Kadoya et al. (2009) implemented citizen science presence data in MaxEnt to predict the potential for invasion by B. terrestris in Japan. These studies underscore the utility of models produced by MaxEnt for bumble bee species policy and planning purposes.

In an effort to update and increase the understanding of statewide bumble bee species presence in Texas, and to inform the planning and implementation of state conservation actions, this study’s aims were to:

(1) Compile statewide bumble bee presence records from novel data sources (citizen science repositories and recent field studies) and natural history collections to produce an updated database of known bumble bee presence in Texas.

(2) Create SDMs using MaxEnt to illustrate the potential ranges of common bumble bee species in Texas.

(3) Identify conservation target areas that are likely to contain multiple declining bumble bee species.

Materials & Methods

Species presence data

Bumble bee presence data were gathered from natural history museum collections, citizen science repositories, and recent field collection data. These were compiled into one database, available at https://figshare.com/articles/Texas_Bombus_Records/4805761, for mapping and species distribution modeling purposes.

Previously compiled natural history collection records (Warriner, 2012; L Richardson, 2017, unpublished data; see also https://figshare.com/articles/Texas_Bombus_Records/4805761) included data from the following institutions: Elm Fork Natural Heritage Museum, Texas A & M University Insect Collection, Texas Memorial Museum, Illinois Natural History Survey Insect Collection, Cornell University Insect Collection, Florida State University Collection of Arthropods, Mississippi State University Entomological Museum, K.C. Emerson Entomology Museum, Purdue Entomological Research Collection, Smithsonian Natural History Museum, University of Arkansas Arthropod Museum, University of Georgia Collection of Arthropods, University of Michigan Museum of Zoology, University of Minnesota Insect Collection, American Museum of Natural History, Canadian National Collection, Essig Museum of Entomology, Los Angeles County Museum, Ohio State University, Bohart Museum of Entomology, University of Colorado Museum of Natural History, California State Collection of Arthropods, University of California Riverside Entomology Research Museum, Lyman Entomological Collection, Bee Biology and Systematics Laboratory, USGS Patuxent Wildlife Research Center, and the Yale Peabody Museum. When possible, records lacking geographic coordinates were georeferenced using Google Earth; record localities given as county names were georeferenced to coordinates corresponding to the county seat.

Two citizen scientist repositories, iNaturalist and the Texas Bumblebees Facebook page (hereafter referred to as “Texas Bumblebees”), were also mined for presence records. Only records that included photographs that could be identified to species, as well as location data, were incorporated into the modeling database. The iNaturalist platform allows contributors to include locality coordinates with their submissions; when available, these coordinates were included in the presence database. Records lacking coordinates, as well as all submissions from Texas Bumblebees, were georeferenced based on locality information provided by the observer using Google Earth. Several citizen scientist records were also directly submitted as photographs to J Beckham. Species determinations followed Williams et al. (2008) and Colla, Richardson & Williams (2011). Additionally, field collection data were gathered from recent research projects performed by J Beckham and M Warriner in 2010–2016 (Beckham et al., 2016; J Beckham, 2016, unpublished data; see also https://figshare.com/articles/Texas_Bombus_Records/4805761).

Correcting for sampling bias

A fundamental assumption in MaxEnt is that the presence data upon which models are built have been acquired from random sampling across the study area. However, museum and citizen science data are not typically collected in a systematic fashion, resulting in spatially biased data. Running MaxEnt without correcting for this sampling bias results in overfitting of models to reflect survey effort rather than actual species distributions. When sample size is sufficient, spatial filtering is recommended to minimize errors of omission and commission in MaxEnt resulting from using spatially biased presence data (Kramer-Schadt et al., 2013). Additionally, because MaxEnt uses presence-only data to produce SDMs, background points are selected to contrast against the presence locations. By default, MaxEnt assigns every pixel the same probability of being selected as a background point. Selecting background points so that they carry the same bias as (unfiltered) presence data aids in the production of more accurate distribution models (Phillips et al., 2009; Merow, Smith & Silander, 2013). In order to achieve this end, a bias grid with cell values weighted to reflect non-uniform sampling effort (following Elith, Kearney & Phillips, 2010) can be implemented in MaxEnt for biased apriori background point selection.

The presence data included in the present study were biased towards heavily populated areas in Texas, especially the Dallas-Fort Worth metroplex, San Antonio, Houston, and Austin. To account for this bias, presence data were spatially filtered and a bias grid was produced to modify the random selection of background points using tools from the SDMtoolbox (Brown, 2014) implemented in ArcMap 10.2.2 (http://support.esri.com/en/download/2093; ESRI, Redlands, WA, USA). Presence data were spatially filtered using the SDMtoolbox rarefy occurrence data for SDMs tool that reduces spatial autocorrelation by removing duplicate occurrence points and reduces occurrence records to a single point within a specified area; for our data we used a 15-km2 resolution. The bias grid was created using the SDMtoolbox Gaussian kernel density of sampling localities tool loaded with presence points from all species; a sampling bias distance of 30 km was chosen. The resulting spatially filtered presence data and bias file were utilized in MaxEnt.

Preparation of environmental variables

Environmental layers of 19 derived bioclimatic variables were downloaded from the WorldClim global climate database (Hijmans et al., 2005, available at www.worldclim.org) at a resolution of 10 min (∼344 km2) as ESRI grids. This resolution was chosen because it allowed us to include lower resolution data that had been georeferenced to county. All rasters were clipped to match the boundaries of the state of Texas using the extract by mask tool in ArcMap 10.2.2; the cartographic boundary file for the mask was downloaded from https://www.census.gov/geo/maps-data/data/tiger-cart-boundary.html and a shapefile of Texas was produced and projected to WGS 1984. All clipped environmental rasters were converted to ASCII files in ArcMap for use in MaxEnt.

It is recommended that correlation be minimized between predictor variables implemented in MaxEnt, as would be done for a traditional statistical model (Merow, Smith & Silander, 2013). A Pearson’s correlation analysis was performed on all 19 layers using the correlation and summary stats tool available for ArcMap in the SDMtoolbox, and pairs of environmental layers with correlation coefficients >0.80 were identified (Table S1). Variables were ranked prior to analyses according to perceived ecological importance, with derived variables (BIO02, BIO03, BIO04, BIO07, and BIO15) being given lowest ranking. Variables pertaining to the warmest and wettest months and quarters were ranked above similar variables involving coldest and driest months and quarters given that bumble bees are very generally more active in warmer, wetter months. When variable importance could not be intuited based on the aforementioned criteria, ranking was based upon the arbitrary WorldClim code. The lower-ranking predictor of each correlated pair was removed, resulting in a total of ten environmental predictors being maintained for species distribution analyses (Table 1).

Table 1 WorldClim bioclimatic environmental predictor variables used for MaxEnt analyses.

Variables are presented in the order of perceived environmental importance that was used to eliminate correlated variables, with derived variables (BIO02, BIO03, BIO04, BIO07, and BIO15) being given lowest ranking. Variables pertaining to the warmest and wettest months and quarters were ranked above similar variables involving coldest and driest months and quarters given that bumble bees are generally more active in warmer, wetter months. When variable importance could not be intuited based on the aforementioned criteria, ranking was based upon the arbitrary WorldClim code. Variables included in further analyses are highlighted in bold. When applicable, the correlate upon which removal of a variable was based, as well as the Pearson’s correlation coefficient (r), is indicated. A Pearson’s correlation coefficient of 0.8 or higher was used as the cutoff for variable removal. The full results of the Pearson’s correlation analysis are available in Table S1.

Rank for Pearson’s correlation analysis	WorldClim code	Environmental predictor variable	Basis for removal from analysis (correlate and Pearson’s correlation coefficient, if applicable)	
1	BIO01	Annual mean temperature		
2	BIO12	Annual precipitation		
3	BIO10	Mean temperature of warmest quarter		
4	BIO08	Mean temperature of wettest quarter		
5	BIO09	Mean temperature of driest quarter		
6	BIO11	Mean temperature of coldest quarter		
7	BIO5	Max temperature of warmest month	BIO10; 0.92216	
8	BIO6	Minimum temperature of coldest month	BIO11; 0.97781	
9	BIO13	Precipitation of wettest month	BIO12; 0.88484	
10	BIO14	Precipitation of driest month	BIO12; 0.9619	
11	BIO18	Precipitation of warmest quarter		
12	BIO19	Precipitation of coldest quarter	BIO12; 0.9529	
13	BIO16	Precipitation of wettest quarter	BIO12; 0.9028	
14	BIO17	Precipitation of driest quarter	BIO12; 0.96877	
15	BIO02	Mean diurnal range (mean of monthly (max temp −min temp))		
16	BIO03	Isothermality (BIO2/BIO7) (*100)		
17	BIO04	Temperature seasonality (standard deviation*100)		
18	BIO07	Temperature annual range (BIO5-BIO6)	BIO11; −0.85041	
19	BIO15	Precipitation seasonality (Coefficient of variation)	BIO12; −0.81672	

Production of Species Distribution Models (SDMs)

Input data for species distribution modeling in MaxEnt consisted of the aforementioned spatially filtered presence data (occurrence records) and the 10 environmental layers (predictor variables). The Gaussian kernel density bias file was also loaded to alter selection of background points to reflect sampling bias. MaxEnt was run for 100 replicates for each species with default settings, except as follows: jackknifing was selected to measure variable importance, 30% test percentage was implemented, and 5,000 max iterations were allowed. The average logistic output over 100 replicates was calculated in MaxEnt for each species, and resulting SDM rasters were visualized in ArcMap 10.2.2. MaxEnt’s logistic output assigns each pixel in the study area a value between 0 and 1, and is the closest approximation of the probability of species presence (Elith et al., 2011). The default MaxEnt species prevalence value (an estimate of the probability of species presence at “typical” presence locations for the target species) was used to transform the raw MaxEnt output into the logistic output for each SDM. The reasoning behind using MaxEnt’s default value of 0.5, which can be interpreted as there being a 50% chance of the target species being present in a suitable location, is that the actual species prevalence values are rarely known and hard to approximate. Thus, calculated probability values from MaxEnt’s logistic output are interpreted not as the probability of species occurrence, but instead as the probability of suitable habitat being present for the target species. For detailed mathematical explanations of MaxEnt’s logistic output see Elith et al. (2011) and Phillips, Anderson & Schapire (2006).

SDM raster processing and analyses

For visualization and analyses purposes, SDM rasters were reclassified using the reclassify tool in ArcMap’s Spatial Analyst toolbox from the continuous scale of 0–1 probability calculated in MaxEnt to a categorical 0–2 scale, with 0 being low probability of suitable habitat being present (<0.25), 1 being medium probability (0.25–0.49), and 2 being high probability (>0.5). These rasters were then displayed with the underlying EPA Level III ecoregions (spatial layer downloaded from https://www.epa.gov/eco-research/ecoregion-download-files-state-region-6; see Fig. 1). Identifying the ecoregions associated with each bumble bee species will help inform decisions regarding the planning and implementation of conservation plans. Furthermore, because funds and manpower dedicated to conservation can be limited, we identified areas where suitable habitat was likely to be present for two of the declining species found in Texas (B. fraternus and B. pensylvanicus). In order to do this, the combine tool in ArcMap’s Spatial Analyst toolbox was used to overlay the reclassified B. fraternus and B. pensylvanicus rasters, and cells with high (>0.5) probabilities of suitable habitat for both were identified.

Figure 1 EPA Level III ecoregions in the state of Texas (shapefile downloaded from https://www.epa.gov/eco-research/ecoregion-download-files-state-region-6).

Model validation

The accuracy of each SDM was evaluated using values of the area under the Receiver Operating Characteristic (ROC) curve, or AUC value, plotted and calculated in MaxEnt based on the training and test data. These values convey the predictive performance of the model as compared to a selection of random background points, and can be interpreted as the probability that any presence site is ranked higher in terms of habitat suitability for the target species than a random background site (Merow, Smith & Silander, 2013). An AUC value of 0.5 reflects a model that is no better than random, while an AUC of 1 would be considered a “perfect” model. We trained each model on 70% of the presence data points, and then tested it with the remaining 30% of the data, resulting in both training and test AUC values for each target species’ SDM.

We then tested the null hypothesis that the average test AUC values generated for each species were significantly different from those predicted by a null model, as described in Raes & Ter Steege (2007). We generated a null distribution for each species by first randomly drawing 999 sets of random collection localities from the study area; the number of points in each set was equal to the corresponding number of spatially filtered presence points used for the target species’ SDM. A model was then produced in MaxEnt for each set of points, with all settings identical to those used to produce the actual SDMs. The test AUC was calculated for each null SDM, and a null distribution of the 999 test AUCs for the random replicates was produced. The actual model’s average test AUC was then compared to the upper 95% confidence limit of the null distribution to test the null hypothesis (alpha level of 0.05). Random draws were performed in R (version 3.3.3, https://cran.r-project.org/index.html, 2017) using the RandomPoints function contained in the dismo package (Hijmans et al., 2017), and confidence limits were calculated in SAS (version 9.4; SAS Institute, Cary, NC, USA).

Results

Species presence

A total of 3,580 Texas records spanning 1897–2016 were compiled from museum collections, citizen science repositories and recent fieldwork (Table 2). These data included 747 previously unpublished citizen science records from the past ten years that were recorded in iNaturalist and Texas Bumblebees (Fig. 2). Eleven species were identified from these records, including one specimen of B. appositus Cresson, 1878, five specimens of B. morrisoni Cresson, 1878, and two specimens of B. nevadensis Cresson, 1874, which had not previously been documented in the list of Texas bumble bee species compiled by Warriner (2012). The updated species list for Texas, along with numbers of specimens records, is as follows: B. appositus (n = 1), B. auricomus (n = 13), B. bimaculatus (n = 5), B. fervidus (n = 4), B. fraternus (n = 239), B. griseocollis (n = 108), B. impatiens (n = 173), B. morrisoni (n = 5), B. nevadensis (n = 2), B. pensylvanicus (n = 3,010), and B. variabilis (n = 20). Notably, a subset of 314 B. pensylvanicus records were also logged as presenting the sonorus morphology.

Table 2 Sources of bumble bee presence data.

Data source (time period)	Number of records	
Museum collections (1897–2012)	2,105	
iNaturalist (2007–2016)	377	
Texas bumble bees facebook (2007–2016)	370	
Field data (2010–2016)	728	
Total	3,580	

Figure 2 Novel presence data (n = 747) compiled from the citizen science repositories iNaturalist and Texas Bumblebees Facebook page.

Included species: B. auricomus (n = 11), B. fraternus (n = 48), B. griseocollis (n = 60), B. impatiens (n = 29), B. pensylvanicus (n = 557), and B. pensylvanicus sonorus (n = 42). EPA Level III ecoregions are shown for reference (a, Arizona/New Mexico Mountains; b, Chihuahuan Deserts; c, High Plains; d, Southwestern Tablelands; e, Central Great Plains; f, Cross Timbers; g, Edwards Plateau; h, Southern Texas Plains; i, Texas Blackland Prairies; j, East Central Texas Plains; k, Western Gulf Coastal Plain; l, South Central Texas Plains).

Species distribution models

Of the twelve species represented by the data, four species comprised 99% of the records, and so we only modeled the distributions for these most abundant species: B. fraternus, B. griseocollis, B. impatiens, and B. pensylvanicus. Sample sizes for the other species known from Texas were deemed insufficient for spatial filtering and modeling purposes. The logistic outputs of SDMs produced in MaxEnt, as well as the reclassified visualizations of these models, for B. fraternus, B. griseocollis, B. impatiens, and B. pensylvanicus are shown in Figs. 3–6, respectively.

The contribution of each bioclimatic variable to each SDM varied across the four species, as did the most important variables for each model. Annual precipitation (BIO12) was the top contributor to SDMs for B. griseocollis (71.6%), B. impatiens (73.8%), and B. pensylvanicus (20.3%). Isothermality (BIO03), calculated as [[mean diurnal temperature range/mean annual temperature]*100] was the top contributor to the SDM for B. fraternus (51% contribution), and was the second and fifth most important variable for B. griseocollis (8.5%) and B. impatiens (2.1%) SDMs, respectively. The mean temperature of the driest quarter (BIO09) was also a top-five contributor for all models. The top five bioclimatic variables contributing to each SDM, as well as their percent contributions, are presented in Table 3.

Figure 3 Modeling results for B. fraternus.

(A) Mean logistic output from MaxEnt, averaged over 100 replicates, showing probability of suitable habitat being present for B. fraternus (average training AUC over 100 runs = 0.826 ± 0.027 and average test AUC 0.745 ± 0.06). Unfiltered species presence data are represented as black dots. (B) Reclassified SDM for B. fraternus in Texas shown with EPA Level III ecoregions outlined (a, Arizona/New Mexico Mountains; b, Chihuahuan Deserts; c, High Plains; d, Southwestern Tablelands; e, Central Great Plains; f, Cross Timbers; g, Edwards Plateau; h, Southern Texas Plains; i, Texas Blackland Prairies; j, East Central Texas Plains; k, Western Gulf Coastal Plain; l, South Central Texas Plains). Suitable habitat for B. fraternus is likely across the northern half of Texas, with highest modeled probabilities found in the Cross Timbers, Texas Blackland Prairies, and East Central Texas Plains ecoregions.

Figure 4 Modeling results for B. griseocollis.

(A) Mean logistic output from MaxEnt, averaged over 100 replicates, showing probability of suitable habitat being present for B. griseocollis (average training AUC over 100 runs = 0.903 ± 0.02 and average test AUC 0.866 ± 0.04). Unfiltered species presence data points are represented as black dots. (B) Reclassified SDM for B. griseocollis in Texas shown with EPA Level III ecoregions outlined (a, Arizona/New Mexico Mountains; b, Chihuahuan Deserts; c, High Plains; d, Southwestern Tablelands; e, Central Great Plains; f, Cross Timbers; g, Edwards Plateau; h, Southern Texas Plains; i, Texas Blackland Prairies; j, East Central Texas Plains; k, Western Gulf Coastal Plain; l, South Central Texas Plains). The eastern half of the state is most likely to contain suitable habitat for B. griseocollis, with the highest probabilities in the northern portions of the Texas Blackland Prairies, East Central Texas Plains, and South Central Plains ecoregions.

Figure 5 Modeling results for B. impatiens.

(A) Mean logistic output from MaxEnt, averaged over 100 replicates, showing probability of suitable habitat being present for B. impatiens (average training AUC over 100 runs = 0.931 ± 0.015 and average test AUC 0.9 ± 0.05). Unfiltered species presence data points are represented as black dots. (B) Reclassified SDM for B. impatiens in Texas shown with EPA Level III ecoregions outlined (a, Arizona/New Mexico Mountains; b, Chihuahuan Deserts; c, High Plains; d, Southwestern Tablelands; e, Central Great Plains; f, Cross Timbers; g, Edwards Plateau; h, Southern Texas Plains; i, Texas Blackland Prairies; j, East Central Texas Plains; k, Western Gulf Coastal Plain; l, South Central Texas Plains). The highest probabilities of suitable habitat modeled for B. impatiens in Texas are in the eastern portion of the state, especially in the South Central Plains ecoregion.

Figure 6 Modeling results for B. pensylvanicus.

(A) Mean logistic output from MaxEnt, averaged over 100 replicates, showing probability of suitable habitat being present for B. pensylvanicus (average training AUC over 100 runs = 0.747 ± 0.012 and average test AUC 0.667 ± 0.03). Unfiltered species presence data are represented as black dots. (B) Reclassified SDM for B. pensylvanicus in Texas shown with EPA Level III ecoregions outlined (a, Arizona/New Mexico Mountains; b, Chihuahuan Deserts; c, High Plains; d, Southwestern Tablelands; e, Central Great Plains; f, Cross Timbers; g, Edwards Plateau; h, Southern Texas Plains; i, Texas Blackland Prairies; j, East Central Texas Plains; k, Western Gulf Coastal Plain; l, South Central Texas Plains). Suitable habitat for this species is likely to be observed in most parts of the state.

To evaluate the validity of each SDM, we calculated both training and test AUC values on 70% and 30% of the presence data, respectively. As expected, test AUC values were lower than training AUCs, but all values were above 0.65. We then tested the hypothesis that the average test AUC for each SDM was significantly different from that of a random null distribution (alpha = 0.05). All average test AUC values from actual SDMs were significantly higher than the values calculated from the null distributions (p < 0.01), and so we concluded that all models were statistically significant. The average training and test AUC values for each SDM, as well as the upper 95% confidence intervals of the null distributions, are contained in Table 3.

According to our results, B. pensylvanicus is both the most common and the most widespread bumble bee species in Texas, representing 84% of the specimens in this study. While it is most likely to be found in the central and eastern portions of the state, there have been specimens recorded in every ecoregion. B. fraternus (∼7% of specimens) is most likely to be found in the northeast portion of the state, but its range may extend to far north and far west Texas. By contrast, the ranges of both B. impatiens (5% of specimens) and B. griseocollis (3% of specimens) are currently limited primarily to the eastern portions of Texas.

Priority areas for actions targeting the conservation of declining species were identified by overlaying the reclassified maps of B. fraternus and B. pensylvanicus in ArcMap to ascertain whether there were areas of the state that carried high probabilities of suitable habitat being present for both species (Fig. 7). This analysis revealed a region of northeast Texas that covers portions of the Cross Timbers, Texas Blackland Prairies, and East Central Texas Plains ecoregions where habitat is likely to be highly suitable for both species.

Table 3 Analysis of species distribution model validity and variable contributions.

The total number of occurrence records (N) and the number of spatially filtered occurrence records used for SDM production (N’) are shown for each species, along with average training and test AUCs for each species distribution model. The upper 95% confidence interval of the null distribution’s test AUC for each species is also shown for model validation purposes; note that p < 0.01 for all species, meaning that the modeled distributions are significantly different from the simulated null distributions. Lastly, top variables contributing to the SDM of each bumble bee species are included, along with percent contribution of each.

Species	N	N′	Avg. training AUC ± S.D. (number of training points in each replicate)	Avg. test AUC ± S.D. (number of test points in each replicate)	Upper 95% C.I. of null distribution’s test AUC (p-value of comparison of actual SDM vs. null)	Top five variables contributing to model (percent contribution)	
B. fraternus	239	69	0.826 ± 0.027 (49)	0.745 ± 0.06 (20)	0.6102 (p < 0.01)	bio03 (51%); bio09 (11%); bio08 (10.2%); bio18 (9.2%); bio01 (4.4%)	
B. griseocollis	108	39	0.903 ± 0.02 (28)	0.866 ± 0.04 (11)	0.6572 (p < 0.01)	bio12 (71.6%); bio03 (8.5%); bio08 (5.6%); bio09 (4.2%); bio04 (2.8%)	
B. impatiens	173	24	0.931 ± 0.015 (17)	0.9 ± 0.05 (7)	0.6958 (p < 0.01)	bio12 (73.8%); bio09 (8.6%); bio08 (7.6%); bio01 (4.1%); bio03 (2.1%)	
B. pensylvanicus	3,010	322	0.747 ± 0.012 (226)	0.667 ± 0.03 (96)	0.5217 (p < 0.01)	bio12 (20.3%); bio09 (15.2%); bio11 (12.9%); bio02 (12.5%); bio04 (11.7%)	

Figure 7 Target areas for Texas bumble bee conservation.

Areas where there is a high (>0.5) probability of suitable habitat being present for the declining B. fraternus and B. pensylvanicus species are highlighted, and EPA Level III ecoregions are outlined (a, Arizona/New Mexico Mountains; b, Chihuahuan Deserts; c, High Plains; d, Southwestern Tablelands; e, Central Great Plains; f, Cross Timbers; g, Edwards Plateau; h, Southern Texas Plains; i, Texas Blackland Prairies; j, East Central Texas Plains; k, Western Gulf Coastal Plain; l, South Central Texas Plains). Conservation target areas were calculated using the combine tool in ArcMap’s Spatial Analyst Toolbox; reclassified distributions of B. fraternus and B. pensylvanicus were overlaid in implementation of this tool to achieve results. A region of northeast Texas including portions of the Cross Timbers, Texas Blackland Prairies, and East Central Texas Plains ecoregions were identified as target areas for conservation actions.

Discussion

The species presence records documented in this study included three new, albeit historically very rare, species from the list of Texas bumble bees previously published by Warriner (2012): B. appositus (n = 1), B. morrisoni (n = 5), and B. nevadensis (n = 2). These records were retrieved from the database of museum records managed by Leif Richardson (Williams et al., 2014; L Richardson, 2017, unpublished data; see also https://figshare.com/articles/Texas_Bombus_Records/4805761), and records were validated with museums of origin. As such, we have revised the species list of Texas bumble bees to include 11 species.

In comparison to the Texas portion of the range map presented in Williams et al. (2014), our model for B. fraternus was somewhat contracted, lacking the high likelihood of presence predicted by Williams et al. across the Western Gulf Coastal Plain, and throughout the Southwestern Tablelands and High Plains. By contrast, we identified novel presence records for both B. impatiens and B. griseocollis in the East Central Texas Plains and South Central Plains, resulting in a shift of the western edges of these species’ Texas ranges from those predicted by Williams et al. (2014). Because B. griseocollis populations are considered stable across its range (Colla & Packer, 2008; Colla & Ratti, 2010), and B. impatiens populations are considered stable (Cameron et al., 2011) or possibly expanding (Colla & Packer, 2008), our results suggest that Texas populations should be monitored for further evidence of range expansion for these species. The model of the range of B. pensylvanicus presented by Williams et al. (2014) is similar to our model, which is virtually statewide. Had we included presence records from the neighboring states of Louisiana, Oklahoma, and New Mexico, our models, especially the model for B. fraternus, might have been extended further into border ecoregions. Future work should include further improvement of SDMs for these species that include records from neighboring states.

One drawback to presence-only species distribution modeling is that models can be biased towards areas that contain more presence records (Kramer-Schadt et al., 2013). In the case of our data, presence records from areas with larger human populations (i.e., the Dallas-Fort Worth metroplex, Austin, San Antonio, and Houston) were more common than less populated areas of the state. In order to address this problem, presence records were spatially filtered and a Gaussian bias file was implemented in MaxEnt. However, these techniques did not completely resolve the issue of sampling bias, and individual species distributions were still biased towards highly populated areas. Conversely, SDMs for areas that were not well sampled, such as the Southwestern Tablelands and High Plains ecoregions located in north central Texas, predicted low or moderate likelihood of any bumble bee species presence. Syfert, Smith & Coomes (2013) similarly showed that, save an intentional sampling strategy across the study area, SDMs will be somewhat biased towards clustered data. Thus, we consider our SDMs to be conservative, and suggest cautious interpretation of species presence in marginally sampled ecoregions. Targeted field surveys in these areas are needed to achieve both a more uniform statewide bumble bee sampling effort and less clustered data for SDMs.

Of the ten predictor variables included in the SDMs, we interpret our results to mean that annual precipitation, isothermality, and the mean temperature of the driest quarter are the most important bioclimatic factors for predicting bumble bee habitat suitability in Texas. Precipitation was the top variable contributing to the SDMS for three of the four species studied (B. griseocollis, B. impatiens, and B. pensylvanicus), and isothermality was the top contributor to the B. fraternus model, as well as a top-five contributor to two other SDMs. The mean temperature of the driest quarter was one of the top five environmental predictors for all SDMs. As global climate change threatens to disrupt temperature and precipitation patterns, the current models may change drastically. Furthermore, though these environmental factors are undoubtedly important for bumble bee habitat, it should be noted that other factors for which data were not available may also influence the suitability of habitat, such as pesticide use, available nesting and foraging sites, and habitat connectivity.

While multiple factors have been implicated in bumble bee declines, habitat loss, often associated with urban sprawl and agricultural intensification, is considered to be a leading cause (Kearns, Inouye & Waser, 1998; Winfree et al., 2009). Bumble bee conservation initiatives designed to curb these declines should include actions that establish, enhance, and maintain habitat. The declining B. fraternus and B. pensylvanicus persist in northeast Texas, where a substantial area of native rangeland remains (Beckham et al., 2016); across the state, native rangelands comprise approximately 63% of the state’s nonfederal rural land (U.S. Department of Agriculture, 2013), and encouraging the conservation of these lands amidst rapid population growth and urbanization may prove beneficial to bumble bees. Additionally, practices such as targeted agri-environment schemes that increase floral biodiversity have been shown to increase bumble bee diversity and abundance in agricultural systems (Pywell et al., 2006; Carvell et al., 2007; Carvell et al., 2011) and, within urban environments, green spaces such as parks and community gardens can provide valuable habitats for bumble bees in otherwise unfavorable landscapes (McFrederick & LeBuhn, 2006; Ahrne, Bengtsson & Elmqvist, 2009; Goulson et al., 2010; Beckham, 2016; J Beckham, 2016, unpublished data; see also https://figshare.com/articles/Texas_Bombus_Records/4805761). In Texas, where over 95% of land is privately owned (Texas Land Trends, 2014), the conservation of bumble bee species will require a multifaceted approach, including public education and outreach, as well as working with municipalities and private landowners to create and maintain pollinator habitat.

The results of our multi-species analyses suggest starting points for conservation programs when funding and manpower is limited. The target area includes portions of Texas’ Cross Timbers, Blackland Prairies, and East Central Plains, where suitable habitat for both B. fraternus and B. pensylvanicus is highly likely to co-occur. Notably, this target area fully contains the Dallas-Fort Worth metroplex, a heavily urbanized area that saw a decrease in working lands acreage of almost 316,000 acres, and whose human population grew by about 2.3 million people, from 1997 to 2012 (Texas Land Trends, 2015). However, the conservation target area also contains surrounding rural areas across the region, and so our results have underscored the need for conservation strategies that will address both urban and rural environments.

Conclusions

Our study has updated the knowledge of presence and distribution of the four most common bumble bee species in Texas, to include B. griseocollis, B. impatiens, and the declining B. fraternus and B. pensylvanicus. By combining novel data from vetted citizen science records reported over the last ten years (i.e., 2007–2016) and data from recent field surveys (2010–2016), with previously compiled museum specimen records, the most complete database of Texas bumble bee records available has been established. These records have resulted in a revised species list for the state that includes three species previously not known to Texas. Additionally, while continental range maps have previously been modeled for bumble bee species found in Texas, we have produced fine-scale SDMs using MaxEnt and identified priority areas for bumble bee conservation efforts that will be practical for state-level conservation planning.

Supplemental Information

Table S1 Results of Pearson’s correlation analysis of Worldclim bioclimatic predictor variables

Pearson correlation coefficients (r) are shown in the table for each pair of predictor variables. Indices >0.8 are highlighted in red. Analysis was performed in ArcMap using the correlation and summary stats tool from the SDMtoolbox.

Click here for additional data file.

Thank you to Leif Richardson and Michael Warriner, who provided historical bumble bee records from the state of Texas, and to the many individuals and institutions who contributed to those databases. We also thank the following individuals for providing insight into individual historic records: John Ascher and Hadel Go regarding records from the American Museum of Natural History, Karen Wright regarding records from the Texas A&M University Insect Collection, and Doug Yanega regarding records from the Essig Museum of Entomology at Berkeley. Additionally, thank you to the many citizen scientist volunteers who contributed hundreds of bumble bee records to iNaturalist, Texas Bumblebees, and directly to this project.

Additional Information and Declarations

Competing Interests

Author Contributions

Data Availability

The authors declare there are no competing interests.

Jessica L. Beckham conceived and designed the experiments, performed the experiments, analyzed the data, contributed reagents/materials/analysis tools, wrote the paper, prepared figures and/or tables, reviewed drafts of the paper.

Samuel Atkinson conceived and designed the experiments, performed the experiments, analyzed the data, contributed reagents/materials/analysis tools, wrote the paper, reviewed drafts of the paper.

The following information was supplied regarding data availability:

Beckham, Jessica (2017): Texas Bombus Records. figshare.

https://doi.org/10.6084/m9.figshare.4805761.v1.

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
