# Peer review of "An updated understanding of Texas bumble bee (Hymenoptera: Apidae) species presence and potential distributions in Texas, USA"

_PeerJ, doi:10.7717/peerj.3612_

## Round 0.1 · original submission · Minor Revisions

Both reviewers found the paper made a great contribution but pointed out some minor revisions needed.

·

Basic reporting

This manuscript is well-written and appropriately referenced. All figures and tables are relevant and well executed, although Table 1 could be improved with the suggestion outlined below. The raw data used for analyses have been made available and contain sufficient detail.

Experimental design

The experimental design of this paper is excellent, and more impressively, very well described. With the level of detail provided, there is no doubt that the analytical methods employed here could be easily replicated.

Validity of the findings

The results of these analyses are directly applicable to the research aims of the manuscript, and the inferences are solid.

Additional comments

Presence of B. vagans in Texas: Of the new species records in the state, that of B. vagans seems particularly unusual and, for that reason, it might warrant a small bit of discussion. The other three species have ranges somewhat close to Texas and might occasionally appear as vagrants within the Texas borders, yet B. vagans has not to my knowledge been reported in areas near the state. A vagrant would have to travel quite far to land within the Texas borders! There are records in neighboring New Mexico of B. cockerelli, a species that may or may not be con-specific with B vagans (Yanega, 2013; Williams, et al., 2014). Did the morphology of the B. vagans specimen match that of the typical B. vagans or did it conform to the description of B. cockerelli?
Specific suggestions:
Line 62 vs Line 584: I believe the correct capitalization is Labougle
Line 69: The Kopec, 2017 citation is not listed in the references section, and it should be added. This likely refers to the grey literature report “Pollinators in Peril”, produced by the Center for Biological Diversity.
Lines 204–207 and Table 1: Because correlated variables were removed based on “perceived ecological importance”, it would be valuable to see the removed variables and their correlates, as well as the reasoning behind the choices made, included in the table.
Line 183: Include the manufacturer and location of ArcMap 10.2.2 here (ESRI, Redlands, CA). Also, this is listed in the References (Line 507; as ArcGIS, rather than ArcMap) as a full citation without being referenced in the text.
Lines 231, 240–241, 265: For consistency, italicize the names of the tools used (e.g., Reclassify), as you have done previously (e.g., Lines 202–203).
Line 239: Because there are more than two species that are considered declining throughout their ranges and also occur in Texas, the word “both” should be replaced with a more general term, such as “two of”.
Line 275 vs Line 280: There is a small discrepancy in the number of B. vagans reported. Line 275 reports n=2; Line 280 reports n=1. As there is only one record in the attached data file, I think that Line 275 needs to be revised to report a single specimen.
Lines 688–690: This reference (Tilman, et al., 2001) is not included within the body of the manuscript.
Table 3: Should B. pensylvanicus N’= 322? There were 226 training points and 96 test points used.
Review References:
Williams, P. H., R. W. Thorp, L. L. Richardson, and S. R. Colla. 2014. Bumble Bees of North America: An Identification Guide. Princeton University Press, Princeton, NJ.
Yanega, D. 2013. The status of Cockerell's bumblebee, Bombus (Pyrobombus) cockerelli Franklin, 1913 (Hymenoptera: Apidae). Southwestern Entomologist 38:517–522.

Reviewer 2 ·

Basic reporting

Beckham & Atkinson present an investigation on the distribution of Texas bumble bees using georeferenced natural history collection data, citizen science data, and standard insect collections. With the data, they constructed species distribution models using the MaxEnt software program and Worldclim bioclimatic data set. The paper is well written, and cites sufficient peer-reviewed literature throughout. The figures are well made, except for its color scheme. The authors should consider using a color scheme that does not use reds and greens in differentiating ecoregions. This approach potentially excludes those with red-green color blindness. Also, the authors do not provide the raw data they used in their study. The authors need to consider providing their dataset along with the manuscript. The paper is direct- and examines bumble bee habitat suitability in Texas, a U.S. state with few studies on bumble bee conservation status studies.

Experimental design

The authors conduct original research on bumble bee habitat suitability in Texas and examine their distribution in the context of EPA ecoregions. Their goals are well defined and meaningful as they seek to identify areas in Texas that should be targeted for monitoring and conservation. Their species distribution modelling approach is standard- and in line with current approaches using MaxEnt. Their methods are well explained. However, the authors need to provide their raw data in the form of an appendix.

Validity of the findings

The authors present meaningful results, however they should expand on the significance of the different bioclimatic variables used in their study in the context of their target species. Their study would be greatly improved if they discussed why they think the variables that came out to be significant to model constructions are in fact significant in the context of their target species’ biology. Addressing the significance of their bioclimatic variables used in the SDMs will enhance our understanding of how the species may respond to continued climate change over the next century.

Additional comments

Line 198: Change WGS_1984 to WGS 1984

Line 344: Complete citation of Williams et al. with the year of publication.

Line 372: I suggest that you visually characterize the relationship between the top three climate variables of your four bumble bee species. For example, you identified that precipitation, isothermality, and mean temperature of the driest quarter are significant predictors. What is the distribution of these values relative to the distribution of your target species? Do they inhabit wetter/drier, warmer/cooler, and seasonal environments? Visualizing the climate data will aide in the understanding of how the species are distributed across their bioclimatic niche. Furthermore, it will be useful in estimating how continued climate change may impact habitat suitability. While SDMs are useful in identifying habitat suitability- they don’t clearly capture the relationship between climate and species distribution. Thus, consider visualizing the relationship between significant bioclimatic variables and the distribution of your target species.

Figure 1: Consider changing your figure color schemes to accommodate individuals with red/green color blindness (e.g. can’t tell difference between High Plains and Edwards Plateau)

Figures 3 – 7: You identify EPA ecoregions where each of your target bumble bees are estimated to have high HS in your figure legend. However, it is inconvenient to return to Figure 1 to determine which ecoregion is associated with each polygon. I suggest you use some identifier (e.g., lower case letters) to identify which ecoregion is which for each species map- and place that identifier in your figure legend as well: e.g., “ a) Texas Blackland Prairies, b) East Central Texas Plains, and c) South Central Plains ecoregions.” (From Figure 4)

---

## Round 0.2 · accepted · Accept

Many thanks for your thorough work on the responses and revised ms